# Catecholamines Induce Left Ventricular Subclinical Systolic Dysfunction: A Speckle-Tracking Echocardiography Study

**DOI:** 10.3390/cancers11030318

**Published:** 2019-03-06

**Authors:** Jan Kvasnička, Tomáš Zelinka, Ondřej Petrák, Ján Rosa, Branislav Štrauch, Zuzana Krátká, Tomáš Indra, Alice Markvartová, Jiří Widimský, Robert Holaj

**Affiliations:** 13rd Department of Medicine, Centre for Hypertension, General University Hospital and 1st Faculty of Medicine, Charles University in Prague, Ovocný trh 5, 116 36 Prague 1, Czech Republic; jan.kvasnicka3@vfn.cz (J.K.); tzeli@lf1.cuni.cz (T.Z.); ondrej.petrak@vfn.cz (O.P.); jan.rosa@vfn.cz (J.R.); branislav.strauch@vfn.cz (B.Š.); zuzana.kratka@vfn.cz (Z.K.); alice.vrankova@vfn.cz (A.M.); jwidi@lf1.cuni.cz (J.W.J.); 2Department of Nephrology, General University Hospital and 1st Faculty of Medicine, Charles University in Prague, Ovocný trh 5, 116 36 Prague 1, Czech Republic; tomas.indra@vfn.cz

**Keywords:** pheochromocytoma, catecholamine, global longitudinal strain, speckle-tracking echocardiography, subclinical systolic dysfunction

## Abstract

*Background*: Pheochromocytomas (PHEO) are tumors arising from chromaffin cells from the adrenal medulla, having the ability to produce, metabolize and secrete catecholamines. The overproduction of catecholamines leads by many mechanisms to the impairment in the left ventricle (LV) function, however, endocardial measurement of systolic function did not find any differences between patients with PHEO and essential hypertension (EH). The aim of the study was to investigate whether global longitudinal strain (GLS) derived from speckle-tracking echocardiography can detect catecholamine-induced subclinical impairments in systolic function. *Methods*: We analyzed 17 patients (10 females and seven males) with PHEO and 18 patients (nine females and nine males) with EH. The groups did not differ in age or in 24-h blood pressure values. *Results*: The patients with PHEO did not differ in echocardiographic parameters including LV ejection fraction compared to the EH patients (0.69 ± 0.04 vs. 0.71 ± 0.05; NS), nevertheless, in spackle-tracking analysis, the patients with PHEO displayed significantly lower GLS than the EH patients (−14.8 ± 1.5 vs. −17.8 ± 1.7; *p* < 0.001). *Conclusions*: Patients with PHEO have a lower magnitude of GLS than the patients with EH, suggesting that catecholamines induce a subclinical decline in LV systolic function.

## 1. Introduction

Pheochromocytomas (PHEO) and functional paragangliomas (PGLs) are rare and mostly non-metastatic tumors originating from chromaffin cells either from the adrenal medulla (PHEO) or from the sympathetic nervous system–associated chromaffin tissue (PGLs) [1]. The prevalence of PHEO and PGLs in non-selected population of patients with arterial hypertension is between 0.2 and 0.6% [2,3] and the prevalence of PHEO is higher than the prevalence of PGLs, when 80 to 85% of chromaffin-cell tumors are PHEO, whereas 15 to 20% are PGLs [4]. Due to the higher age of the population and smaller tumor sizes at diagnosis, the incidence has increased in recent years [5].

These tumors have the ability to produce, metabolize, and secrete catecholamines. Catecholamines produced by the tumor cells are responsible for a large variety of signs, in particular paroxysmal effects, such as headache, sweating, palpitations, and hypertension because of their effect on hemodynamics and metabolism [4,6]. In vitro [7] and in vivo studies [8] showed that catecholamines influence vascular wall growth and remodeling, independently of their hemodynamic impact. In general, patients with pheochromocytoma have a higher risk of cardiovascular complications (even life-threatening like arrhythmias, heart failure and myocardial infarction), than patients with essential hypertension (EH) [9]. The aforementioned heart failure may be manifested by a decrease in the ejection fraction (EF) or, in some patients, by a transient left ventricle (LV) dysfunction due to the so-called catecholamine-induced myocarditis, also called pheochromocytoma-associated catecholamine cardiomyopathy [10]. Adrenalectomy also leads to an improvement of LV mass in patients with PHEO in contrast to the impairment of this parameter in EH patients [11]. A reduction of LV EF or even heart failure are signs of already developed clinical impairment. We therefore focused on the detection of subclinical impairment before the onset of cardiac damage.

In recent years, global longitudinal strain (GLS) derived from two-dimensional speckle-tracking echocardiography seems to be a better parameter for evaluating LV systolic performance including myocardial motion and longitudinal deformation than LV EF [12]. GLS can also detect LV systolic impairment already in the preclinical stage, when EF remains in normal range [13]. Recently, GLS has been used for the assessments of LV subclinical systolic function in many indications. In clinical practice, it is most often the evaluation of various forms of LV hypertrophy such as hypertrophic cardiomyopathy, amyloidosis [14] or primary aldosteronism [15] and evaluation of cardiotoxicity in patients with oncological diseases undergoing chemotherapy [16]. Therefore, we designed a prospective study to detect catecholamines-induced myocardial impairment of LV systolic function in patients with PHEO already in the subclinical stage.

## 2. Results

### 2.1. Characteristic of Groups

The final group included seventeen patients with a diagnosis of PHEO (11 subjects with adrenergic phenotype and six subjects with noradrenergic phenotype), aged 28 to 67 years (10 females and seven males) and eighteen patients (nine females and nine males) with a diagnosis of EH. The patient subgroups do not significantly differ in age, body mass index, in presumptive duration of disease or in heart rate and blood pressure values measured casually or using 24-h ambulatory monitoring (ABPM). 

Thirteen patients with PHEO (76%) had a history of sustained hypertension and used at least one antihypertensive drug. Four patients with PHEO (24%) had developed only paroxysmal symptoms in the history and displayed normal blood pressure levels during measurements in the hospital. On the contrary, two patients with PHEO (12%) showed repeatedly very high blood pressure levels. The other patients with PHEO showed only a mild form of hypertension. The average values of heart rate in patients with PHEO were only about +7 mmHg higher than those in patients with EH. Nevertheless, this slight difference did not achieve statistical significance. The patients with EH used a higher number of antihypertensive drugs before switching to the treatment with α-blockers and/or slow-release verapamil than the patients with PHEO (*p* < 0.01) (Table 1). Significantly higher proportion of EH patients were treated by β-blockers (*p* < 0.01), calcium channel blockers (*p* < 0.01), and diuretics (*p* < 0.05). Four patients with PHEO had diabetes (two of them were on insulin and three of them were on oral antidiabetic drugs) and seven patients in both groups were treated for dyslipidemia (Table 2).

### 2.2. Laboratory Results

The patient subgroups did not differ in lipid parameters, in plasma creatinine or in creatinine clearance. As expected, all endocrine-related laboratory values in patients with PHEO (fasting plasma glucose, plasma metanephrines, normetanephrines) were higher than in patients with EH (Table 3).

### 2.3. Echocardiography Parameters

The patient subgroups did not differ in the LV and left atrial dimensions, LV mass indexes or Doppler-derived indexes characterizing diastolic function (Table 4). 

When evaluating systolic function, the two groups did not differ in LV EF (0.69 ± 0.04 in the PHEO group vs. 0.71 ± 0.05 in the EH group, *p* = 0.25), nevertheless, in the speckle analysis, a significantly lower magnitude of GLS was found in patients with PHEO compared to those with EH. 

The patients with PHEO displayed significantly lower strain than those with EH in all three views including: apical two-chamber view (−14.9 ± 1.6% in the PHEO group vs. −18.2 ± 2.1% in the EH group, *p* < 0.001), apical long axis view (−15.0 ± 1.7% in the PHEO group vs. −18.0 ± 1.9% in the EH group, *p* < 0.001), apical four-chamber view (−14.5 ± 1.4% in the PHEO group vs. −17.8 ± 1.7% in the EH group, *p* < 0.001), and GLS (−14.8 ± 1.5% in the PHEO group vs. −17.8 ± 1.7% in the EH group, *p* < 0.001, Figure 1). Comparing the individual LV segments, patients with PHEO showed a significantly reduced peak longitudinal strain in all segments (apical, mid-ventricular and basal, *p* < 0.001) compared to patients with EH (Table 5). 

## 3. Discussion

Our results demonstrate that the patients with PHEO display a lower magnitude of GLS than patients with EH, although they display the same hemodynamic parameters and no difference in LVEF. Our study therefore indicates that the overproduction of catecholamines in patients with PHEO may cause subclinical LV systolic impairment.

A conventional approach to the assessment of LV systolic function usually involves measurement of LVEF and endocardial fractional shortening. Both of these methods are derived from endocardial movement without considering myocardium deformation [17]. However, an evaluation using the EF cannot detect LV affection in hypertrophic patients and thus distinguish these patients from healthy controls [18,19].

In contrast to the conventional methods of measuring LV from endocardial movement, methods taking myocardium deformation into consideration, such as mid-wall endocardial fractional shortening or speckle-tracking echocardiography, have the advantage of being able to detect systolic function impairments with a higher degree of sensitivity [12]. For example, in hypertensive patients, an impairment of LV longitudinal strain and geometric changes (such as concentric remodeling or hypertrophy) occur prior to a decrease in LVEF [20], and increased afterload-related cardiomyocyte hypertrophy and collagen deposition in the extracellular matrix may cause the deterioration in strain in hypertensive myocardium [21].

It is well known that catecholamine overproduction has an adverse effect on cardiac structure [22]. Catecholamine stimulates cell growth and cardiomyocyte hypertrophy, which may lead to cardiac wall thickening and LV mass increase. In addition, in animal studies, catecholamine infusion has been shown to induce cardiac hypertrophy and myocardial interstitial fibrosis and scarring in both left and right ventricles [23]. In our previous studies, we found that the patients with PHEO displayed higher LVMI than patients with essential hypertension in echocardiography and that adrenalectomy led to a reduction of cardiovascular remodeling [11]. Catecholamine-induced cardiomyocyte hypertrophy, elevated cardiac wall thickness and collagen deposition in the extracellular matrix may explain the decrease in GLS in patients with PHEO in the current study. With a more contemporary tool, cardiac magnetic resonance imaging, Ferreira at al. [24] demonstrated that patients with PHEO had broader extent of myocardial fibrosis and myocardial dysfunction than patients with EH and elevated LV mass, and cardiac fibrosis improved after the removal of catecholamine excess (after adrenalectomy). Taken together, these findings indicate that patients with PHEO had a more severe cardiac fibrosis. This reduction in GLS is well known from patients with hypertrophic cardiomyopathy in whom reductions in longitudinal strain may be found prior to the reduction in EF [12].

An important factor is the direct toxic effect of catecholamines on the myocardium. Long-term high levels of catecholamines lead to β-adrenergic receptor downregulation. This degrades the function of the myofibers and gradually leads to their necrosis [25]. A similar mechanism, where β-adrenergic receptors are stimulated, occurs in stress cardiomyopathy, also referred to as Takotsubo cardiomyopathy [26]. Also, patients with PHEO may develop Takotsubo-like cardiomyopathy [27] due to the overproduction of catecholamines or develop a different form of another cardiac dysfunction (inverted Takotsubo cardiomyopathy and diffuse hypokinesis of LV) [10,28], which may often be transient [29]. Known cardiotoxic effects have been also observed in different types of chemotherapeutics used in the treatment of oncological diseases (although not mediated through β-adrenergic receptors), and GLS is used for an early detection of this cardiotoxicity [30]. It is therefore suggested that the decrease in GLS is related to the direct toxic effect of catecholamines on the myocardial muscle fibers through β-adrenergic receptors.

Another mechanism that may also play a role in the acceleration of cardiovascular hypertrophy is the higher fasting plasma glucose concentration in subjects with PHEO [31]. Asymptomatic patients with type 2 diabetes mellitus have a significant reduction in GLS, which is associated with a worse prognosis in these groups of patients [32]. Similarly, patients with type 2 diabetes mellitus have lower GLS after ST-segment elevation myocardial infarction [33] than non-diabetic patients. 

Finally, a chronic inflammatory process may also lead to vascular damage [34,35]. In our previous study, we showed that chronic catecholamine excess in subjects with PHEO was accompanied by an increase in inflammatory markers, which was reversed by the tumor removal [36]. The decline of GLS is well documented in patients with the systemic inflammatory response syndrome and the magnitude of decline of GLS is related to the prognosis of these patients [37], which can be also related to the results of our work.

There are several limitations to this study. First, the number of patients was relatively small, which prevented finding any association between GLS and catecholamine overproduction. However, this is the first study to demonstrate subclinical systolic functional changes in patients with PHEO using speckle-tracking echocardiography. Further large-scale studies are needed to confirm the link between the magnitude of GLS and the overproduction of catecholamines. Secondly, we tried to match the group of subjects with PHEO with the EH group as closely as possible. This is, however, an elusive goal, because the overproduction of catecholamines leads not only to hypertension and weight loss but also to abnormalities in glucose metabolism. This makes the exact matching of the two groups unachievable. Therefore, the possible impact of diabetes on the magnitude of GLS in subject with PHEO cannot be excluded. On the other hand, EH patients had definitely higher atherogenic risk profiles, namely longer duration of hypertension, higher 24 h ABPM systolic blood pressure and higher body weight which may counteract the GLS differences between the two groups. Thirdly, the frequency of various antihypertensive drugs was not identical in the two groups of patients. Intervention studies in EH patients found that a therapy with drugs affecting the renin–angiotensin aldosterone system and calcium channel blockers can have a superior effect on the regression of LV hypertrophy than a therapy with diuretics and β-blockers independently of BP lowering. In our study, the proportions of EH patients on angiotensin-converting enzyme inhibitors or calcium channel blockers therapy were higher than those of PHEO patients (56% vs. 29% and 78% vs. 29%, respectively). Fourthly, postoperative speckle-tracking data were not available at the time of the study. Therefore, we could not resolve whether the impaired subclinical systolic function was reversible or not. A follow-up study involving postoperative findings of speckle-tracking analysis is under way.

## 4. Materials and Methods

Patients were recruited from a cohort of almost 1100 patients investigated for severe or resistant hypertension and for suspected secondary hypertension at our tertiary hospital-based Centre for Hypertension at the 3rd Department of Medicine, General University Hospital and 1st Faculty of Medicine, Charles University in Prague between November 2015 and October 2018. Each participant provided his/her written informed consent, and the study protocol was approved by the local Ethics Committee which took place during the grant approval (on 21 May 2015, code 20/15).

The diagnosis of PHEO was newly confirmed in 35 patients during the aforementioned period, which is about 3% rate in this preselected population. The diagnosis of PHEO was based on elevated plasma metanephrines and normetanephrines above the upper reference limit, and positive finding of adrenal tumor on computed tomography or magnetic resonance imaging. After examination all subjects underwent surgical removal of the tumor, and the diagnosis was confirmed histo-pathologically.

Ten patients were not enrolled due to the poor quality of echocardiography images or impossibility of GLS determination and seven due to significant comorbidities, including coronary atherosclerosis, atrial fibrillation or cardiac dysfunction for reasons other than PHEO. One patient was excluded for persistent overproduction of catecholamines after surgical removal because of the generalization of metastatic PHEO.

The control group of patients with essential hypertension (EH) was composed of the same prospective cohort as for the PHEO patients, on the basis of matching age, gender, body mass index, office and 24 h systolic blood pressure (BP). The patients were selected, after exclusion of the main forms of secondary hypertension (primary aldosteronism, PHEO, Cushing syndrome, renal parenchymal disease, renovascular hypertension), non-compliance or drug-induced hypertension. The subjects were considered hypertensive or pre-hypertensive when their clinic BP, an average of 3 sphygmomanometric measurements performed on 3 separate days, was ≥140/90 mmHg or ≥130/80 mmHg, respectively [38]. Chronic antihypertensive therapy was discontinued at least 2 weeks before admission, and patients were switched to the treatment with α-blockers and/or slow-release verapamil. Diabetes mellitus was defined as medication with oral antidiabetic drugs or repeated fasting glucose levels of >7.0 mmol/L [39]. There were two insulin-dependent patients in the PHEO group and none in the control group. All subjects with dyslipidemia (total plasma cholesterol ≥5.0 mmol/L or low-density cholesterol ≥3.0 mmol/L or high-density lipoprotein cholesterol ≤1.0 mmol/L in men and ≤1.2 mmol/L in women or triglycerides ≥1.7 mmol/L) were on a diet and received lipid-lowering therapy [40]. All patients were examined during a short three-day hospitalization.

### 4.1. BP Measurement

Casual blood pressure was measured using an oscillometric device (Omron M6, Shimogyo-ku, Kyoto, Japan). The measurement was made in a silent, quiet room with the patient’s arm situated at the heart level and on chronic antihypertensive treatment during the first ambulatory visit, prior to switching to the treatment with α-blockers and/or slow-release verapamil. Blood pressure was measured three times in sitting position after five minutes of rest. The resulting value of causal systolic and diastolic blood pressure was calculated as the average from the second and third measurements. The patient’s 24-h blood pressure was measured during their stay in the hospital using an oscillometric device (SpaceLabs 90207, SpaceLabs Medical, Redmond, WA, USA) already on switched medication.

### 4.2. Laboratory

Plasma-fractioned metanephrines (metanephrine and normetanephrine) were quantified by liquid chromatography with electrochemical detection (Agilent 1100; Agilent Technologies, Wilmington, DE, USA) in the Laboratory for Endocrinology and Metabolism at the 3rd Department of Medicine, General University Hospital and 1st Faculty of Medicine, Charles University in Prague [41].

Blood biochemistry, including sodium, potassium, urea, creatinine, total cholesterol, low-density lipoprotein cholesterol, high-density lipoprotein cholesterol, triglycerides, and plasma glucose, was analyzed using a multianalyzer (Modular SWA; Roche Diagnostics, Basel, Switzerland) in the Institute of Medical Biochemistry and Laboratory Diagnostics of the General University Hospital and 1st Faculty of Medicine, Charles University in Prague. Creatinine clearance was calculated using the Cockcroft–Gault equation.

### 4.3. Echocardiography

M-mode, 2-dimensional, Doppler and speckle tracking echocardiography were performed according to a standard protocol on Vivid 9 ultrasound system (GE Healthcare, Chicago, IL, USA). The records were analyzed offline using the EchoPAC working station (v.113, Advanced Analysis Technologies; GE Healthcare) by one cardiologist (J.K.) blinded to participants final diagnoses due to at least a fourteen-day period for the analysis of plasmatic metanephrines. M-mode images of the left ventricle at the mitral valve tip were obtained, guided by 2-dimensional parasternal long-axis and short-axis view, with the subjects lying down in the left lateral decubitus position at end-expiration. The LV end-diastolic (LVED) diameter, interventricular septum (IVS) thickness and LV posterior wall (LVPW) thickness were measured at the end of diastole and relative wall thickness (RWT) was measured with the formula 2 × LVPW thickness/LVED according to the recommendations of the American Society of Echocardiography and the European Association of Cardiovascular Imaging [42]. 

The LVED index was calculated as the LVED diameter indexed to the body surface area in square meters (LVED diameter/body surface area). LV mass estimation using American Society of Echocardiography convention was used [43]: LV mass (grams) = 0.8 × 1.04 × [(LVED diameter + IVS thickness + LVPW thickness)^3^ − (LVED diameter)^3^] + 0.6 (with diameters in centimeters). Two variants of LV mass indexing were used: to the body surface area in square meters and to the 2.7th power of height in meters. The LV EF was measured by the biplane method of disks (modified Simpson’s rule) according to the last published recommendations [42]. Before the speckle-tracking analysis was performed, the image quality, frame rate and foreshortening were optimized. The speckle-tracking analysis was performed by automated detection of endocardial border after manually defining the basal and apical points of the LV myocardium. If necessary, a manual adjustment was applied. The seventeen ventricular segment model was obtained from three projections: apical four-chamber view, two-chamber view and apical long-axis view and then the GLS was computed as the mean of peak longitudinal strain values from each of these segments according to consensus of American Society of Echocardiography and European Association of Echocardiography endorsed by the Japanese Society of Echocardiography (Figure 2) [44]. As recommended, patients were excluded if tracking was insufficient in more than one segment because of not clear visualization or artefacts [45]. If tracking in only one segment was unsuccessful, this segment was discarded and not used when calculating the GLS. The mid-wall GLS and also peak longitudinal strain in individual segments were evaluated. Individual segments were unified like basal, mid-ventricular and apical for simplification. The normal range of GLS using GE Healthcare system was −18.0 to −21.5% ± 3.7% [45].

### 4.4. Statistical Analysis

Data were analyzed using the Stata 13.5 program (StataCorp LP, College Station, TX, USA). Differences between the two groups (PHEO and EH) were analyzed with the help of the χ^2^ test for categoric data and with the help of non-paired t-test for normal distribution of variables for the two patient groups. Depending on the normality/nonnormality of the distributions of particular variables, the results were given as mean ± SD values or median values (interquartile range). A *p*-values of <0.05 were considered statistically significant.

## 5. Conclusions

In conclusion, the patients with PHEO revealed lower magnitudes of GLS than the patients with EH. This finding is possibly caused by catecholamine-induced subclinical decline in LV systolic function, nevertheless, the link between the magnitude of GLS and the overproduction of catecholamines has not been proved in this study. At this stage, we can only express a suspicion of the diagnosis of PHEO in hypertensive patients based on measured lower magnitudes of GLS during routine echocardiographic examination.

## Figures and Tables

**Figure 1 cancers-11-00318-f001:**
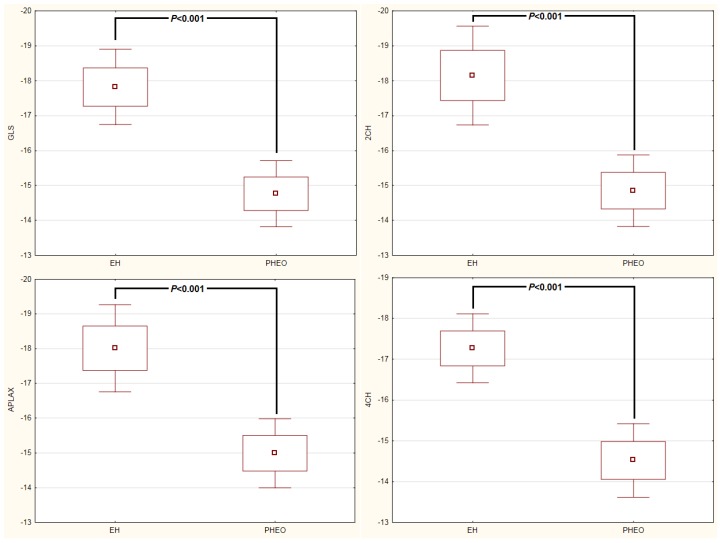
Speckle-tracking analysis in patients with pheochromocytoma and essential hypertension. The patients with pheochromocytoma showed significantly lower global longitudinal strain (GLS), strain in apical two-chamber view (2CH), strain in apical long axis view (APLAX), and strain in apical four-chamber view (4CH).

**Figure 2 cancers-11-00318-f002:**
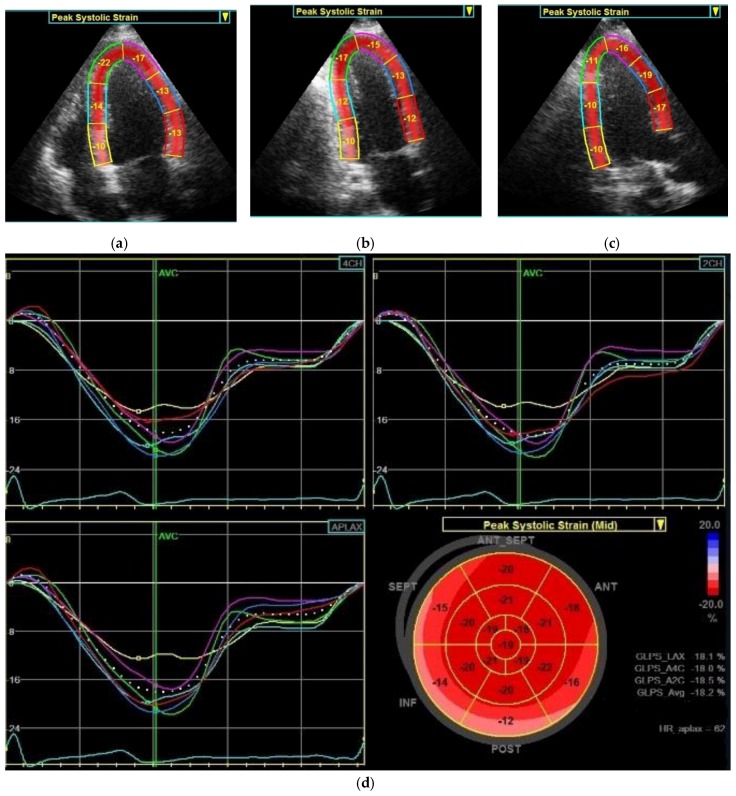
Distribution of individual left ventricular segments in which peak systolic strain is analyzed in apical four-chamber view (**a**), apical two-chamber view (**b**) and apical long-axis view (**c**). The resulting peak systolic strain-expressing curves in individual segments that correspond to the color designation of the segments in images a-c in a patient with essential hypertension (**d**) and in a patient with pheochromocytoma (**e**). The GLS (or GLPS) is calculated for the whole LV from each segment peak systolic strain and is expressed as the LV seventeen-segment model also called “bull eye” which is shown at the bottom right of images d-e. GLPS, global longitudinal peak strain; 4CH, apical four-chamber view, 2CH, apical two-chamber view; APLAX, apical long-axis view; MID, mid-wall; AVC, aortic valve closure; ANT-SEPT, anterior-septal; ANT, anterior; LAT, lateral; POST, posterior; INF, inferior; SEPT, septal; HR, heart rate

**Table 1 cancers-11-00318-t001:** Clinical characteristic of the study population.

Clinical Characteristic	PHEO (*n* = 17)	EH (*n* = 18)	*p*-Value
Age (years)	50 ± 11	49 ± 6	NS
Gender: F/M (% female)	10/7 (58%)	9/9 (50%)	NS
Height (cm)	170 ± 8	173 ± 7	NS
Weight (kg)	82 ± 14	88 ± 11	NS
Body mass index (kg/m^2^)	29 ± 5	30 ± 4	NS
Systolic office BP (mmHg)	141 ± 13	140 ± 8	NS
Diastolic office BP (mmHg)	88 ± 6	89 ± 5	NS
Heart Rate office (BPM)	81 ± 9	74 ± 8	NS
24 h ABPM systolic BP (mmHg)	127 ± 9	132 ± 8	NS
24 h ABPM diastolic BP (mmHg)	76 ± 7	80 ± 5	NS
24 h ABPM Heart Rate (BPM)	77 ± 10	71 ± 6	NS
Number of used antihypertensive drugs	1.5 ± 1.1	3.6 ± 1.4	<0.001
Manifestation of symptoms (years)	5.8 ± 3.4	6.7 ± 3.6	NS

Variables are shown as means ± SD, or absolute values and relative values in percent. PHEO, pheochromocytoma; EH, essential hypertension; BP, blood pressure; BPM, beats per minute; ABPM, ambulatory blood pressure monitoring; NS, non-significant.

**Table 2 cancers-11-00318-t002:** Use of antihypertensive, antidiabetic and lipid-lowering drugs in the study population.

Antihypertensive, Antidiabetic and Lipid-Lowering Drugs	PHEO (*n* = 17)	EH (*n* = 18)	*p*-Value
Diuretics [*n* (%)]	3 (18)	10 (56)	<0.05
β-blockers [*n* (%)]	3 (18)	11 (61)	<0.01
Calcium channel blockers [*n* (%)]	5 (29)	14 (78)	<0.01
Angiotensin-converting enzyme inhibitors [*n* (%)]	5 (29)	10 (56)	NS
Angiotensin receptor blockers [*n* (%)]	2 (12)	7 (39)	NS
α-blockers [*n* (%)]	4 (24)	2 (11)	NS
Central agonists [*n* (%)]	3 (18)	6 (33)	NS
Aldosterone antagonists [*n* (%)]	1 (6)	4 (22)	NS
Statins [*n* (%)]	7 (41)	7 (39)	NS
Insulin [*n* (%)]	2 (12)	0 (0)	NS
Oral antidiabetic drugs [*n* (%)]	3 (18)	0 (0)	NS

Values are presented in absolute numbers (in percents). PHEO, pheochromocytoma; EH, essential hypertension; NS, non-significant.

**Table 3 cancers-11-00318-t003:** Laboratory data of the study population.

Laboratory Data	PHEO (*n* = 17)	EH (*n* = 18)	*p*-Value
Plasma creatinine (µmol/L)	69 ± 12	75 ± 12	NS
Creatinine clearance (mL/min)	135 ± 34	119 ± 25	NS
Plasma cholesterol (mmol/L)	4.4 ± 0.5	4.8 ± 0.5	NS
HDL cholesterol (mmol/L)	1.5 ± 0.3	1.5 ± 0.3	NS
LDL cholesterol (mmol/L)	2.4 ± 0.5	2.5 ± 0.5	NS
Triglycerides (mmol/L)	1.2 ± 0.5	1.4 ± 0.5	NS
Fasting plasma glucose (mmol/L)	6.0 ± 0.9	5.2 ± 0.5	<0.05
Plasma metanephrines (nmol/L)	4.87 ± 4.30	0.16 ± 0.09	<0.01
Plasma normetanephrines (nmol/L)	13.65 ± 13.80	0.27 ± 0.12	<0.05

Variables are shown as means ± S.D.; PHEO, pheochromocytoma; EH, essential hypertension; HDL, high-density lipoprotein; LDL, low-density lipoprotein; NS, non-significant.

**Table 4 cancers-11-00318-t004:** Echocardiographic parameters and Doppler-derived indexes of the study population.

Echocardiographic Parameters	PHEO (*n* = 17)	EH (*n* = 18)	*p*-Value
IVS (mm)	9.7 ± 1.6	9.6 ± 1.1	NS
LVED (mm)	49.6 ± 4.7	49.3 ± 3.1	NS
LVES (mm)	30.3 ± 2.6	29.1 ± 2.9	NS
PWT (mm)	9.6 ± 1.6	9.8 ± 1.1	NS
RWT	0.39 ± 0.05	0.40 ± 0.05	NS
LA (mm)	38.2 ± 5.1	37.0 ± 2.9	NS
LVMi/BSA (g/m^2^)	91.2 ± 23.3	86.4 ± 16.2	NS
LVMi (g/m^2.7^)	42.2 ± 12.1	40.3 ± 9.1	NS
LVEF	0.69 ± 0.04	0.71 ± 0.05	NS
E/A	1.04 ± 0.30	1.06 ± 0.25	NS
E/e’	8.5 ± 1.9	8.9 ± 1.6	NS

Variables are shown as means ± SD; LVEF, left ventricle ejection fraction; IVS, interventricular septum; LVED, left ventricle end-diastolic diameter; LVES, left ventricle end-systolic diameter; PWT, posterior wall thickness; RWT, relative wall thickness; LVMi/BSA, left ventricular mass index to the body surface area; LVMi, left ventricular mass index to the 2.7th power of height in meters; LA, left atrium; E/e’, Pulsed-Wave Doppler/Tissue Doppler Imaging ratio of E wave velocity, NS, non-significant.

**Table 5 cancers-11-00318-t005:** Longitudinal strain parameters of the study population.

Longitudinal Strain Parameters	PHEO (*n* = 17)	EH (*n* = 18)	*p*-Value
Global LS (%)	−14.8 ± 1.5	−17.8 ± 1.7	<0.001
Basal LV LS (%)	−14.8 ± 2.1	−17.3 ± 2.3	<0.05
Mid-ventricular LV LS (%)	−15.7 ± 1.9	−18.9 ± 2.1	<0.001
Apical LV LS (%)	−16.1 ± 2.6	−19.9 ± 3.9	<0.05

Variables are shown as means ± SD; EF, ejection fraction; GLS, global longitudinal strain, LV LS, left ventricle longitudinal strain.

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
