# Peer review of "Catecholamines Induce Left Ventricular Subclinical Systolic Dysfunction: A Speckle-Tracking Echocardiography Study"

_cancers, 2019, doi:10.3390/cancers11030318_

Round 1
Reviewer 1 Report
In this article, Jan K et.al investigated global longitudinal strain in patients with pheochromocytoma. Pheochromocytoma is a relatively rare diease, so the sample size (n=17) seems to be sufiicient and meaningful. Alotough the impact is not so high as the follow-up data is missing, the authors clearly mentions about limitation and the novelity is good. However some issues need to be addressed.
Clarify the methodology of matching between PHEO and EH.(e.g. propensity score matching etc.)
All patients with PHEO included in this study have almost normal cardiac function. Severity of cardiac impairment of PHEO may be dependent of the duration of catecholamine exposure. How long is it in this tudy?
Author Response
Thank you for your valuable comments.
In this article, Jan Kvasnička et.al investigated global longitudinal strain in patients with pheochromocytoma. Pheochromocytoma is a relatively rare disease, so the sample size (n=17) seems to be sufficient and meaningful. Although the impact is not so high as the follow-up data is missing, the authors clearly mentions about limitation and the novelty is good. However some issues need to be addressed.
Clarify the methodology of matching between PHEO and EH.(e.g. propensity score matching etc.)
The control group of patients with essential hypertension (EH) was composed of the same prospective cohort as the group of PHEO patients, on the basis of matching age, gender, body mass index, and office and 24h systolic blood pressure (BP). We tried to match the group of subjects with PHEO with the EH group as closely as possible. This is, however, an elusive goal, because the overproduction of catecholamines leads not only to hypertension and weight loss but also to abnormalities in glucose metabolism. This makes the exact matching of the two groups impossible. As a result, the possible impact of diabetes on magnitude of GLS in subjects with PHEO cannot be excluded. On the other hand, EH patients had definitely higher atherogenic risk profiles, namely longer duration of hypertension, higher 24h ABPM systolic blood pressure and higher body weight, which may counteract the GLS differences between the two groups. Unfortunately, propensity score matching or other sophistic statistical method were not used. On the other hand, we do not believe in their additive effect specifically in our study. We added this opinion to the discussion of the new version of the manuscript.
All patients with PHEO included in this study have almost normal cardiac function. Severity of cardiac impairment of PHEO may be dependent of the duration of catecholamine exposure. How long is it in this study?
An estimated duration of catecholamine exposure was 5.8 years. This value is noted as “Manifestation of symptoms” in Table 1 in the new version of the manuscript.
Reviewer 2 Report
In this study, the authors have investigated the occurrence of global longitudinal strain (GLS) in a patient cohort with pheochromocytoma using speckle-tracking echocardiography, and compared the findings to patients with essential hypertension (EH). The authors observed a significant reduction regarding the GLS in patients with pheo compared to EH patients, although the left ventricle ejection fraction was similar. The authors conclude that the pheo associated catecholamines could have induced a subclinical decline in LV systolic function.
The study is original, well-conducted and of clinical interest. Even so, some clarifications and corrections are needed:
The reason for the observed difference in GLS between groups is not established in this study, and the authors should therefore soften their conclusions regarding the proposed mechanism. For example, there is no information regarding dopamine or ACTH production in the pheos et.c. Although it's highly likely that the effects are due to the secreted catecholamines, the authors have not the associated experimental evidence needed to pinpoint the exact pathophysiologic mechanism.
To build on query 1 above, did the authors compare the GLS between patients within the pheo patient cohort? If a correlation would be seen between individual catecholamine levels and/or tumor size and the GLS - this would strengthen their hypothesis that the catecholamines causes the reduction in GLS. I know the amount of patients (n=17) might be to scarce for statistically significant findings, but perhaps a trend can be seen?
How come the manifestation of disease was so long for patients with pheo or paraganglioma (mean 5,8 years)? Is this an estimated value based on the symptomatology, or is it time from diagnosis? If the former statement is true, this must be stated - as an assumed duration is less exact than date of diagnosis. This is important as the authors claim that both groups have similar duration of disease.
The pheo group did not differ from the EH group in terms of 24h ABPM systolic blood pressure or heart rate (Table 1). To me, this is quite surprising given the paroxysmal nature of the hypertensive attacks of a pheo patient. Please comment.
Could the authors please be more specific in terms of the clinical benefits of their results? Since patients with pheo undergo surgery and no postoperative data on pheo patients is presented here, do we know if the phenomenon is irreversible or not? I lack this type of discussion in the manuscript.
Rows 32-33: All PPGL exhibit malignant potential according to the latest WHO classification of Endocrine Tumors (2017). Also, the terminology of "benign" and "malignant" should therefore be replaced with "non-metastatic" and "metastatic".
Figure 2 needs some clarifications to the legend, so that even a non-echocardiography oriented physician could understand the different labels and axes.
Author Response
Thank you for your valuable comments.
In this study, the authors have investigated the occurrence of global longitudinal strain (GLS) in a patient cohort with pheochromocytoma using speckle-tracking echocardiography, and compared the findings to patients with essential hypertension (EH). The authors observed a significant reduction regarding the GLS in patients with pheo compared to EH patients, although the left ventricle ejection fraction was similar. The authors conclude that the pheo associated catecholamines could have induced a subclinical decline in LV systolic function.
The study is original, well-conducted and of clinical interest. Even so, some clarifications and corrections are needed:
The reason for the observed difference in GLS between groups is not established in this study, and the authors should therefore soften their conclusions regarding the proposed mechanism. For example, there is no information regarding dopamine or ACTH production in the pheos et.c. Although it's highly likely that the effects are due to the secreted catecholamines, the authors have not the associated experimental evidence needed to pinpoint the exact pathophysiologic mechanism.
Based on our results, we are not able to postulate pathophysiological mechanisms, of how catecholamines influence on the magnitude of GLS. Some may speculate about deprivation of myocardial contractility due to catecholamine-mediated inflammation, nevertheless, the mediators of the inflammatory response were not estimated in our study. We hope in getting partial answers after an increase of patients in our follow-up. Therefore, we softened our conclusions in the new version of the manuscript.
To build on query 1 above, did the authors compare the GLS between patients within the pheo patient cohort? If a correlation would be seen between individual catecholamine levels and/or tumour size and the GLS - this would strengthen their hypothesis that the catecholamines causes the reduction in GLS. I know the amount of patients (n=17) might be to scarce for statistically significant findings, but perhaps a trend can be seen?
Unfortunately we did not estimate urinary catecholamines in all patients with PHEO. As a result, we are not able to set any correlation between catecholamines and GLS. We were looking for a correlation between GLS and all possible cofounders (metanephrines, tumor size, manifestation of symptoms, 24h or office systolic blood pressure, glycaemia etc.) but no meaningful correlation was found. We pointed out this fact in the discussion in the new version of the manuscript.
How come the manifestation of disease was so long for patients with pheo or paraganglioma (mean 5,8 years)? Is this an estimated value based on the symptomatology, or is it time from diagnosis? If the former statement is true, this must be stated - as an assumed duration is less exact than date of diagnosis. This is important as the authors claim that both groups have similar duration of disease.
The term “Manifestation of disease” refers to the duration of symptoms suspected for the diagnosis of PHEO. For a better understanding, we replaced the term “disease” with the term “symptoms” in Table 1 of the new version of the manuscript.
The pheo group did not differ from the EH group in terms of 24h ABPM systolic blood pressure or heart rate (Table 1). To me, this is quite surprising given the paroxysmal nature of the hypertensive attacks of a pheo patient. Please comment.
Four patients with PHEO (24 %) had developed only paroxysmal symptoms in the history. However, they had normal blood pressure levels during the measurements in the hospital. “Normal” blood pressure levels in these patients were balanced by very high levels repeatedly measured in two other patients in that group. Other patients with PHEO showed only a mild form of hypertension. The average values of heart rate in patients with PHEO were only about +7 mmHg higher than in patients with EH. Nevertheless, this slight difference did not achieve statistical significance. We added this information in the new version of the manuscript.
Could the authors please be more specific in terms of the clinical benefits of their results? Since patients with pheo undergo surgery and no postoperative data on pheo patients is presented here, do we know if the phenomenon is irreversible or not? I lack this type of discussion in the manuscript.
Postoperative speckle-tracking data were not available in this study, and therefore we could not speculate whether the impaired subclinical systolic function was reversible or not. Further studies with speckle-tracking analysis before and after tumour removal are needed for this purpose. At present, we can express a suspicion of a diagnosis of PHEO in hypertensive patients based on measured lower magnitude of GLS during routine echocardiographic examination. We added this opinion to the discussion of the new version of the manuscript.
Rows 32-33: All PPGL exhibit malignant potential according to the latest WHO classification of Endocrine Tumours (2017). Also, the terminology of "benign" and "malignant" should therefore be replaced with "non-metastatic" and "metastatic".
To your recommendation and according to WHO classification of Endocrine Tumors (2017), the terminology of "benign" and "malignant" was replaced by "non-metastatic" and "metastatic” in the new version of the manuscript.
Figure 2 needs some clarifications to the legend, so that even a non-echocardiography oriented physician could understand the different labels and axes.
For a better understanding of the speckle-tracking analysis, we added small pictures of LV in individual predefined echocardiographic projections (4CH, APLAX and 2CH views).
Round 2
Reviewer 1 Report
The authors have addressed my questions.